A literature survey of shapelet quality measures for time series classification

Li Teng 1
Guo Xiaodong 1 gxd@sdu.edu.cn
Ji Cun 2 jicun@sdnu.edu.cn
1 Shandong University , Jinan , China
2 Shandong Normal University , Jinan , China
Coelho Paulo Jorge
Electronic publication date: 2025 Aug 14
Publication date: 2025
Volume: 11
Electronic Location ID: e3115
Received 2025 Mar 20; Accepted 2025 Jul 21
Copyright: © 2025 Li et al.
Copyright year: 2025
Copyright holder: Li et al.
License: This is an open access article distributed under the terms of the Creative Commons Attribution License, which permits unrestricted use, distribution, reproduction and adaptation in any medium and for any purpose provided that it is properly attributed. For attribution, the original author(s), title, publication source (PeerJ Computer Science) and either DOI or URL of the article must be cited.
License URL: https://creativecommons.org/licenses/by/4.0/

Keywords: Classification, Data mining, Quality measures, Shapelet, Time series

Funding: Innovation Methods Work Special Project 2020IM020100 Natural Science Foundation of Shandong Province ZR2020QF112 This work was supported by the Innovation Methods Work Special Project under Grant 2020IM020100, and the Natural Science Foundation of Shandong Province under Grant ZR2020QF112. The funders had no role in study design, data collection and analysis, decision to publish, or preparation of the manuscript.

==============================
With the rapid development of the Internet of Things, time series classification (TSC) has gained significant attention from researchers due to its applications in various real-world fields, including electroencephalogram/electrocardiogram classification, emotion recognition, and error message detection. To improve classification performance, numerous TSC methods have been proposed in recent years. Among these, shapelet-based TSC methods are particularly notable for their intuitive interpretability. A critical task within these methods is evaluating the quality of candidate shapelets. This paper provides a comprehensive survey of the state-of-the-art measures for assessing shapelet quality. To present a structured overview, we begin by proposing a taxonomy of these measures, followed by a detailed description of each one. We then discuss these measures, highlighting the challenges faced by current research and offering suggestions for future directions. Finally, we summarize the findings of this survey. We hope that this work will serve as a valuable resource for researchers in the field.

Introduction

With the rapid advancement of the Internet of Things, a multitude of sensors are now pervasive across various sectors, continuously collecting data at regular intervals. Typically, these observational values are sampled uniformly over time, resulting in what is known as time series data. Recently, there has been a significant surge in research focused on extracting valuable insights from this temporal information, making time series analysis a prominent area of study. Among the various tasks associated with this field, time series classification (TSC) stands out as particularly crucial (Le et al., 2024; Wei et al., 2024; Liu & Liu, 2018).

The primary objective of TSC is to accurately predict the classes of unlabeled time series data. Currently, TSC arises in a wide range of real-world scenarios (Du et al., 2024; Bhardwaj et al., 2024; Zhao et al., 2024), including but not limited to electroencephalogram/electrocardiogram classification, emotion recognition, and error message detection. Therefore, TSC has attracted significant attention from a large number of researchers (Mohammadi Foumani et al., 2024; Middlehurst, Schäfer & Bagnall, 2024).

In recent years, numerous methods for TSC have been introduced. These methods can be grouped into several categories, including distance-based methods, feature-based methods, interval-based methods, dictionary-based methods, convolution-based methods, deep learning-based methods, hybrid methods and shapelet-based methods (Middlehurst, Schäfer & Bagnall, 2024). Among these, shapelets are discriminative, class-specific subsequences extracted from time series data that best represent key local patterns of a target class (Ji, Wei & Zheng, 2024). Unlike methods that rely on global comparisons (e.g., Dynamic Time Warping (DWT) or Euclidean distance), shapelet-based classification focuses on identifying these localized, interpretable features, which often correspond to critical events or structural characteristics within the time series (Ji et al., 2023). For instance, analogous to distinguishing plant species by examining the angle at which a stem connects to a leaf rather than comparing entire leaves, shapelets isolate pivotal subsequences that capture subtle yet decisive differences between classes. As objectively existing local patterns, shapelets serve as foundational elements that offer meaningful explanations for classification outcomes. Moreover, shapelets can achieve superior accuracy and robustness in noisy or distorted datasets. By focusing on local features rather than global alignments, they mitigate the impact of noise that might obscure subtle discriminative signals in full-sequence comparisons. Consequently, shapelet-based TSC methodologies, owing to their unique capabilities in interpretability, noise resilience, and localized pattern capture, have emerged as a focal point of sustained academic investigation. These approaches have garnered significant traction in both clinical diagnostics (e.g., electrocardiogram (ECG) arrhythmia detection through QT-interval shapelet matching) and industrial predictive maintenance (e.g., equipment anomaly identification via vibration subsequence profiling), etc., (Qu et al., 2024). Shapelet-based methodologies in medical diagnostics. The distance between a shapelet and time series samples of its corresponding class is significantly smaller than those from other classes. Unlike opaque deep learning models that function as “black boxes”, shapelets explicitly correspond to visually identifiable segments of ECG tracings, thereby achieving superior interpretability over state-of-the-art classifiers while maintaining competitive performance in both accuracy and computational efficiency. A representative instance is the shapelet-base framework (Sun & He, 2019), which demonstrates competitive accuracy in anomaly detection for ECG sensor streams compared to state-of-the-art models, coupled with real-time extensibility to accommodate emerging anomaly types.

Shapelet-based methodologies in industry diagnostics. The industrial sector confronts critical challenges in real-time anomaly detection and root cause analysis due to pervasive noise interference within operational sensor data streams. The shapelet-based approach in Wan et al. (2025) universally outperforms other methods across all testing tasks. Compared with the unified domain incremental learning (UDIL) method, the convertible shapelet learning with incremental learning capability (CSLILC) method achieves performance improvements ranging from 19.29% to 47.53%. Notably, CSLILC demonstrates the most substantial improvement over the lp2 method, with enhancements reaching 18.00% to 77.08%.

A critical component of these shapelet-based methods is the evaluation of candidate shapelets’ quality. Over the past few years, numerous measures have been developed to assess shapelet quality. However, to the best of our knowledge, a comprehensive survey of these measures has yet to be conducted. Similar to our objective are surveys in time series classification. In recent years, some researchers have reviewed time series classification methods. Their works can be broadly categorized into two types: comprehensive surveys of time series classification methods, and reviews focusing on specific subcategories within time series classification. The former encompasses multiple facets of time series classification. For example, Bagnall et al. (2017) investigated the six main categories of methods in time series classification. Middlehurst, Schäfer & Bagnall (2024) expanded the research object to the eight main categories of time series classification methods. Ruiz et al. (2021) summarized the eight mainmain categories of methods in multivariate time series classification. The latter focuses exclusively on specific methodologies. Some of these works focus on a categories type of method. For example, Abanda, Mori & Lozano (2019) focused on distance based method for time series. Ismail Fawaz et al. (2019) and Mohammadi Foumani et al. (2024) aimed to review the deep learning methods for time series classification. Some of these works focus on specific classification problems. For example, Gupta et al. (2020) paid close attention to the early classification problem in time series. Theissler et al. (2022) showed solicitude for explainable time series classification problems. Some of these works focus on specially appointed field. For example, Wang et al. (2022) concerned about the time series classification methods for biomedical applications.

Although some of the previous work (Bagnall et al., 2017; Middlehurst, Schäfer & Bagnall, 2024) reviewed the shapelet-based time series classification methods. However, they mainly summarized the different ways of obtaining shapelets. In summary, there has been no review of shapelet quality measures yet. To address this gap, this paper aims to provide an organized overview of the current state-of-the-art shapelet quality measures. We begin by establishing a taxonomy to categorize these measures systematically. Subsequently, each shapelet quality measure will be described in detail. Finally, we conclude with a summary of the methods reviewed and outline potential future research directions in this domain. We hope that this survey serves as a valuable resource for researchers seeking to deepen their understanding of shapelet quality assessment and its impact on time series classification.

Notations and definitions

For the convenience of readers’ understanding, this section introduce notations and definitions.

Definition 1 (Time series). A time series ( T={t1,t2,⋯,tn}) is an ordered sequence of real-valued data points indexed chronologically, where n denotes the series length.

Definition 2 (Datasets). A time series dataset is defined as a set D={(T1,y1),⋯,(Ti,yi),⋯,(Tm,ym)} where each time series Ti is associated with a label yi. |D| is used to denote the number of time series in D.

Definition 3 (Subsequence). A time series subsequence (S) is a contiguous sequence of extracted from a time series. A subsequence of length l starting at position j on time series T can be denoted as S=Tjl={tj,tj+1,⋯,tj+l−1}.

For visually intuitive, Fig. 1 provides a graphical representation of a time series and one subsequence on it. This illustration visually confirms that a subsequence is a contiguous segment of the original time series.

Figure 1 An example of time series and subsequence.

Definition 4 (Shapelet). A shapelet is a discriminative subsequence S that maximally separates time series classes in a dataset D.

Figure 2 illustrates a shapelet-based classification case on ECG data: time series containing subsequences similar to shapelet S are classified as “Hypocalcemia”, whereas those lacking S are deemed “normal”. This exemplifies the class separation utility of this shapelet.

Figure 2 A demo of the class separation utility of shapelet.

Definition 5 (Shapelet quality measure). Shapelet quality measure is a method which can evaluate the class separation utility of subsequence.

Definition 6 (Distance between two subsequences). The distance between two subsequences of the same length is a distance function that takes two time series subsequences of the same length and returns a non-negative value.

The Euclidean distance is used in this survey. The Euclidean distance between S={s1,s2,⋯,…,sl} and S={r1,r2,⋯,…,rl} can be calculated as shown in Eq. (1). It is also applicable to two time series of the same length.

(1) d(T,S)=∑i=1l(si−ri)2.

Definition 7 (Distance from time series to subsequence). The distance from a time series T to a subsequence S is the minimum distance between S and all contiguous subsequences of T having the same length as S.

Survey methodology

To conduct a literature survey of shapelet quality measures for time series classification, we adopted the following steps to select the related paper as shown in Fig. 3: Searching. This step searched the papers from search engines using relevant queries.

Filtering. This step preliminarily filtered out some irrelevant papers through some simple criteria.

Selection. This step selected the final paper which related shapelet quality measures for time series classification.

Figure 3 Illustration of the paper selection process.

Searching

Initially, we formulated comprehensive query terms to retrieve the maximum relevant literature. For this survey, the following databases were systematically searched: Google Scholar, IEEE Xplore Digital Library, ACM Digital Library, Scopus, and Web of Science. These platforms collectively encompass nearly all major computer science engineering research domains through their interdisciplinary indexing and extensive publication coverage. Considering the theme of this survey, the search keywords of query are designed as: “shapelet” AND “time series classification” AND “quality” AND (“evaluate” OR “evaluation” OR “assess” OR “assessment” OR “measurement” OR “measure”). The number of papers fetched from the selected databases are shown in Fig. 3.

Filtering

Two criteria were used to filter out some irrelevant papers: 1. Duplicate papers only require the formal final version. As shown in Fig. 3, 194 papers were filtered out according to this criteria.

2. The language of the paper must be in English. Papers with other languages will be filter out. As shown in Fig. 3, four papers were filter out according to this criteria.

Selection

Finally, we selected the final paper which related shapelet quality measures for time series classification. At first, we selected the 1,036 officially published papers from the 1,224 papers. An officially published paper must be a journal paper, a conference paper, or a book chapter, and it is not retracted. Following by, we have read the full-text of the 1,036 officially published papers. After filtering out 943 papers that only mentioned the names of measures, only 93 papers that formally described methods for evaluating shapelet quality were ultimately selected. Finally, we identified 20 shapelet quality methods from these 93 papers.

Taxonomy of shapelet quality measures

Time series analysis has garnered substantial research interest due to its ubiquitous real-world applications across diverse domains. The strategic selection of evaluation metrics tailored to specific classification scenarios proves critical for optimizing classification accuracy and preserving operational validity. From the perspective of classification tasks, a taxonomy is proposed to provide an organized overview of existing quality measures for candidate shapelets. In the context of shapelets, taxonomy refers to a systematic classification framework that organizes quality evaluation metrics for candidate shapelets based on their applicability to distinct classification scenarios and structural properties, ensuring optimal metric selection aligned with task-specific requirements. As illustrated in Fig. 4, the shapelet quality measures can be categorized into four groups: General measures: These general measures are applicable across all scenarios. As shown in Fig. 4, they include information gain (Ye & Keogh, 2009, 2011), F-statistic (Lines et al., 2012), Kruskal-Wallis (Lines & Bagnall, 2012), Mood’s median (Lines & Bagnall, 2012), Silhouette score (Charane, Ceccarello & Gamper, 2024), F-measure (Xing et al., 2011), cumulative distance (Liu et al., 2016), and importance measure (Ji et al., 2022).

Measures for binary-class: These measures are specifically designed for binary-class time series classification tasks, where only two class labels exist. Measures for binary-class include area under curve (AUC) (Yan & Cao, 2020), distance measures (Chen & Wan, 2023), location measures (Chen & Wan, 2023), Chi-Square statistic (Bock et al., 2018) and gap (Ulanova, Begum & Keogh, 2015).

Measures for multi-class: Tailored for multi-class time series classification tasks, where the number of class labels exceeds two but is limited. Measures for multi-class classification include the derivative of information gain (Bostrom & Bagnall, 2015), coverage ratio (Yang & Jing, 2024), and a combination of F1-score and AUC (Yan & Cao, 2020).

Measures for ordinal classification: In ordinal classification tasks, class labels exhibit an ordinal relationship (Guijo-Rubio et al., 2020). Therefore, the measures in this category necessitate an additional enhancement of order information. Measures for ordinal classification include ordinal Fisher score (Guijo-Rubio et al., 2020), Spearman’s correlation coefficient (Guijo-Rubio et al., 2020), Pearson’s correlation coefficient (Guijo-Rubio et al., 2020), and concentration and dominance of coverage (CD-Cover) (Jing & Yang, 2024).

Figure 4 Taxonomy of shapelet quality measures from the perspectives of classification tasks.

General shapelet quality measures

General shapelet quality measures address universal time series classification problems. This category of methods is universally applicable to all time series classification tasks in principle. This categories of measures mainly includes information gain, F-statistic, Kruskal-Wallis, Mood’s median, Silhouette score, F-measure, cumulative distance and importance measure.

Information gain

Information gain is a widely recognized quality measure for candidate shapelets and has proven to be highly competitive for shapelet-based time series classification methods (Guijo-Rubio et al., 2020). When shapelets were first proposed (Ye & Keogh, 2009), information gain (Shannon, 1948) was adopted as the quality measure. Information gain is calculated based on entropy. For a given dataset D, the entropy e(D) can be computed using the equation:

(2) e(D)=−∑i=1Cpilog⁡pi,

where C represents the number of classes in the dataset, and pi is is the probability that the time series belongs to class i.

A candidate shapelet S with a distance threshold dε can separate the training dataset into two non-overlapping subsets, DL and DR. Time series in D that are farther from S than dε are placed in DR; otherwise, they are assigned to DL. After this separation, the information gain IG can be calculated as follows:

(3) IG=e(D)−|DL||D|e(DL)−|DR||D|e(DR),

where e(D), e(DL), and e(DR) represent the entropy of the respective datasets, and |D|, |DL|, and |DR| denote the number of time series contained in each corresponding dataset.

At the beginning, the information gains of all candidate shapelets were compared collectively. However, this approach may lead to certain classes being underrepresented. Therefore, researchers select shapelets for each class proportionally, rather than relying solely on the magnitude of information gain (Bostrom & Bagnall, 2015; Le et al., 2024). To address over-fitted shapelets, Kramakum, Rakthanmanon & Waiyamai (2018) aggregated information gain of the shapelet candidate with its surrounding. The information gain value is usually similar for an unbalanced dataset (Chen, Fang & Wang, 2024). For this, Chen, Fang & Wang (2024) adopted the information gain ratio to assess the shapelet candidates. The information gain ratio can be obtained as

(4) IG=1−|DL||D|e(DL)−|DR||D|e(DR)e(D)

where e(D), e(DL), e(DR), |D|, |DL|, and |DR| have the same meanings as them in Eq. (3).

Using information gain to evaluate candidate shapelets is very time-consuming (Bostrom & Bagnall, 2017). One reason for this is that the number of candidate shapelets is massive, as any subsequence can be considered a candidate shapelet. Thus, the quality evaluation process can be accelerated by applying various filtering techniques, such as length limits (Hills et al., 2014; Yamaguchi, Ueno & Kashima, 2023), Symbolic Aggregate approXimation (SAX) (Rakthanmanon & Keogh, 2013; Li et al., 2020), key points (Li, Yan & Wu, 2019; Li et al., 2023), and random sampling (Karlsson, Papapetrou & Boström, 2016; Ji et al., 2023). Another reason for the high computational cost is that evaluating a single candidate shapelet involves significant complexity. To address this, some researchers have accelerated the quality evaluation process by reducing measurement complexity through techniques like subsequence distance early abandonment (Ye & Keogh, 2011), admissible entropy pruning (Ye & Keogh, 2011), and reusing computations (Xing et al., 2011; Mueen, Keogh & Young, 2011; Wan et al., 2023). Additionally, some researchers have proposed faster evaluation measures.

F-statistic

At the same time as proposing the shapelet transformation, Lines et al. (2012) used the F-statistic (Mood, 1950) as an alternative quality measure for candidate shapelets. There are three main steps to calculate the F-statistic: 1. By calculating the distances between each time series in the dataset and the candidate shapelet S, a list of distances L=⟨d1,d2,⋯,dn⟩ is obtained.

2. Separate L into several lists L1,…,Li,…,LC based on class membership, such that the distances between all time series that belong to class i are stored in Li.

3. Calculate the F-statistic quality measure as (5) F=∑i=1C(Li¯−L¯)2/(C−1)∑i=1C∑dj∈Li(dj−Li¯)2/(n-C)

where C is the number of classes, n is the total number of time series, Li¯ is the average of the values in the list Li, and L¯ is the overall average of the values in the entire list L.

In this method, the larger the F value, the better the quality of the candidate shapelet.

Kruskal-Wallis

Lines & Bagnall (2012) first used the Kruskal-Wallis test (Kruskal, 1952) to evaluate the quality of candidate shapelets. As a non-parametric test, the Kruskal-Wallis test assesses the quality of candidate shapelets based on the distribution of distances between each time series and the candidate shapelet S. The procedure to obtain the Kruskal-Wallis statistic involves four main steps: 1. First, calculate the distances between each time series and S.

2. Next, assign a rank value to each distance, ordering them from smallest to largest.

3. Then, split the ranks by class membership into sets R1,…,RC.

4. Finally, the Kruskal-Wallis statistic for S can be calculated as follows: (6) KW=12n⋅(n+1)∑i=1C∑rj∈Rirj2|Ri|−3(n+1)

where n is the total number of time series, C is the number of classes, Ri represents the ranks for class i, and |Ri| is the number of time series in class i.

Mood’s median

Lines & Bagnall (2012) first adopted Mood’s median for shapelets. This is another non-parametric test used to assess the quality of candidate shapelets. The procedure to obtain Mood’s median statistic consists of four main steps: 1. Calculate the distances between each time series and the candidate shapelet S, then obtain their median dm.

2. For each class i, record oi1 (the number of distances above dm) and oi2 (the number of distances below dm).

3. For each class i, calculate ei1 (the expected number of distances above the median) and ei2 (the expected number of distances below the median) as follows: (7) ei1=pi⋅∑k=1Cok1

and (8) ei2=pi⋅∑k=1Cok2

where C is the number of classes, and pi is the probability of class i.

4. According to Hills et al. (2014), the final Mood’s median statistic of S can be obtained as: (9) M=∑i=1C∑j=12(oij−eij)2eij.

F-measure

The F-measure (which is also be called Utility in some paper) was adopted by Xing et al. (2011) to assess the shapelet quality. There are four steps to obtain the F-measure of the shapelet candidate S. 1. Obtain the distances between each time series T and S.

2. Calculate the precision of S with the threshold δ as (10) Precision=Nd<δ∧L(S)Nd<δ.

where Nd<δ is the number of time series with a distance less than δ from S, Nd<δ∧L(S) is the number of time series which have the class label with a distance less than δ from S and in the same class as T.

3. Calculate the recall of S with the threshold δ as (11) Recall=Nd<δ∧L(S)NS.

where Nd<δ is the number of time series in the same class as T, Nd<δ∧L(S) is the number of time series which have the class label with a distance less than δ from S and in the same class as T.

4. Get the final F-measure of S as (12) F=2⋅Recall⋅PrecisionRecall+Precision.

Silhouette score

Zhu, Lu & Sun (2015) adopted the Silhouette score (Rousseeuw, 1987) as the clustering indicator and reduced the number of candidate shapelets. Later, the Silhouette score was directly used as a measure to assess the quality of shapelets by Charane, Ceccarello & Gamper (2024). In their work, the Silhouette score is calculated based on whether a time series has the same class label as that of the candidate shapelet or not. The calculation of the Silhouette score for one candidate shapelet S involves mainly two steps: 1. Calculate the distances between each time series T and S.

2. The final Silhouette score of S can be obtained as (13) S=∑T∉Dcd(T,S)|D|−|Dc|−∑T∈Dcd(T,S)|Dc|max(∑T∉Dcd(T,S)|D|−|Dc|,∑T∈Dcd(T,S)|Dc|)

where c is the class label of the time series that contains S, d(T,S) is the distance between T and S, Dc represents the time series from the same class c, D contains all the time series, |Dc| is the number of time series in Dc, |D| is the number of time series in D, and max is used to obtain the maximum value.

Cumulative distance

Liu et al. (2016) selected shapelets for each class through the cumulative distance separately. For class i, the cumulative distance of shapelet candidate S can be calculated as

(14) cd=∑T∈Did(T,S),

where Di is the set of all time series belonging to class i, d(T,S) is the distance between T and S. Finally, the candidate with the smallest cumulative distance is selected as the shapelet.

Importance measure

Unlike the aforementioned measures of evaluating candidates one by one, Ji et al. (2022) utilize an importance measure to evaluate all candidate shapelets simultaneously with the assistance of a random forest. The following six main steps outline how to obtain the importance measure: 1. Calculate the distance between each time series and each candidate shapelet.

2. Construct a random forest based on the calculated distances. In this process, the candidate shapelets are used as features, while the corresponding distances serve as feature values.

3. Calculate the Gini impurity of each node in the random forest using the formula (15) GI=1-∑i=1Cpi2,

where C is the number of classes and pi is the probability of class i.

4. For each decision node, obtain the importance measure as (16) IM=GI-GIl−GIr,

where GI, GIl, and GIr represent the Gini impurity of the current node, the left sub-node, and the right sub-node, respectively.

5. The importance measure of the candidate shapelet IM(S) is the sum of the importance measures of the nodes that use S as a decision condition.

6. Finally, normalize the importance measures.

Shapelet quality measures for binary-class

Binary-class task represents a specialized category of time series classification where the outcome is restricted to either the positive class or the negative class (Ryabko & Mary, 2012; Huang & Yang, 2024). Generally speaking, the distribution of binary classes is often imbalanced. In such scenarios, the effectiveness of common evaluation measures may not be ideal. Shapelet quality measures for binary-class are specifically optimized for these problems, including methods such as AUC, distance measures, location measures, Chi-Square statistic and gap.

Area under curve

Addressing the class imbalance problem, Yan & Cao (2020) adopted the AUC value to evaluate the quality of candidate shapelets in binary-class problems. The calculation of the AUC value for a candidate shapelet S involves three main steps: 1. Calculate the distance between each time series and the candidate shapelet S.

2. Assign a rank value to the distances, ordering them from smallest to largest.

3. Obtain the AUC value of the candidate shapelet as follows: (17) AUC=∑Ti∈DPri−|DP|∗(|DP|+1)2|DP|∗|DN|

where DP contains all the positive time series, |DP| is the number of positive instances, |DN| is the number of negative instances, and ri is the rank value of the time series Ti.

Distance measure

Chen & Wan (2023) designed a distance measure to evaluate candidate shapelets. Their distance measure includes four steps: 1. Generate a subsequence set for each time series Ti through a sliding window of length L, and record the set as set(Ti)=⟨S1i,⋯,Sji,⋯,S|Ti|−L+1i⟩, where |Ti| is the length of Ti, and Sji is a continuous subsequence starting at position j.

2. For each subsequence Sji, calculate the interclass average distance dinter between it and the time series in other categories as (18) dinter=∑Tk∈D−d(Sji,Tk)|D−|,

where D− is the dataset that contains all time series not belonging to the same class as Ti, and |D−| is the number of such time series. In Eq. (18), the term d(Sji,Tk) represents the distance between Sji and the time series Tk. This distance can be calculated as (19) d(Sji,Tk)=minShk∈set(Tk)⁡d(Sji,Shk),

where d(Sji,Shk) is the Euclidean distance between the subsequence Sji and the subsequence Shk.

3. Using a similar method, calculate the intraclass average distance dintra between Sji and the time series in the same category as (20) dintra=∑Tk∈D+d(Sji,Tk)|D+|,

where D+ is the dataset that contains all time series belonging to the same class as Ti, and |D+| is the number of such time series.

4. Based on the interclass average distance dinter and the intraclass average distance dintra, the overall distance measure can be obtained as (21) dm=dinterdintra.

Location measure

Chen & Wan (2023) believe that the shapelets should meet the following two criteria: (1) the best matching subsequences from the same class are near their location; and (2) the best matching subsequences from different classes are far from their location. Based on this, they designed a location measure to assess the candidate shapelets. Their distance measure includes four steps: 1. Generate a subsequence set for each time series Ti through a sliding window of length L, and record the set as set(Ti)=⟨S1i,⋯,Sji,⋯,S|Ti|−L+1i⟩, where |Ti| is the length of Ti, and Sji is a continuous subsequence starting at position j.

2. For each subsequence Sji, calculate the interclass average location difference pinter between it and the best matching subsequences in other categories as (22) pinter=∑Tk∈D−|p(Sji,Tk)−j||D−|,

where D− is the dataset that contains all time series which are not of the same class as Ti, and |D−| is the number of such time series. In Eq. (22), p(Sji,Tk) is the position of the best subsequence of Tk. The best subsequence of Tk is defined as the subsequence that has the smallest Euclidean distance to Sji.

3. Using a similar method, calculate the intraclass average location difference pintra between Sji and time series in the same category as (23) pintra=∑Tk∈D+|p(Sji,Tk)−j||D+|,

where D+ is the dataset that contains all time series which are in the same class as Ti, and |D+| is the number of such time series.

4. Based on the interclass average location difference pinter and the intraclass average location difference pintra, the location measure can be obtained as (24) pm=pinterpintra.

Chi-square statistic

Bock et al. (2018) adopted the Chi-square χ2 statistic to measure shapelet candidates. There are four steps to get the quality of shapelet candidate (S) with the threshold ( δ): 1. Calculate the distances between each time series T and S.

2. Fill in the corresponding values in the contingency table for r Chi-square statistic according the class label and distance. The contingency table for Chi-square statistic is shown as Table 1.

3. Calculating the Chi-square statistic using the value in Table 1. The Chi-square statistic is defined as (25) Tχ2=N(ac-bd)2(a+b)(a+b)(a+d)(b+d)

where a, b, c, and d can be get from Table 1.

4. Obtain the quality of S as (26) quality=1-Fχ2(Tχ2)

where Tχ2 can be get for Eq. (25), Fχ2(⋅) is the cumulative density function of a χ2-distribution.

Table 1 The contingency table for Chi-square statistic.

Class	d(S,T)≤δ	d(S,T)>δ	
Positive class	a	b	
Negative class	d	c	

Gap

Gap is first adopted to assess the quality of shapelet candidates in scenarios without class labels (Ulanova, Begum & Keogh, 2015). It is also suitable for binary classification. The gap can be calculated as

(27) gap=(μ++σ+)−(μ−+σ−)

where μ+ and σ+ denote the average and standard deviation of the distances between all time series in the positive class and the shapelet candidates, μ− and σ− denote the average and standard deviation of the distances between all time series in the negative class and the shapelet candidates.

Shapelet quality measures for multi-class

In multi-class time series classification, the target comprises three or more distinct classes. When selecting shapelets for such problems, a critical trade-off arises: whether to prioritize shapelets that maximize discriminative capability for individual classes or those that enhance overall classification performance (Bostrom & Bagnall, 2015). To address this scenario, shapelet quality measures for multi-class typically incorporate class-specific optimization when evaluating shapelet candidates. Measures in this categories contains the derivative of information gain, coverage ratio, a combination of F1-score and AUC, etc.

Derivative of information gain

The common measures evaluate how well one candidate shapelet splits up all the classes (Bostrom & Bagnall, 2015). However, the selection of shapelets performed by them may not separate a single class. This phenomenon becomes more severe as the number of classes increases. For this, Bostrom & Bagnall (2015) used one vs. all encoding scheme to transform the multi-class into C (C is the class number) binary-class problem. And then, they adopted the derivative of information gain to evaluate the quality of candidate shapelets. The process of obtaining the derivative of information gain is basically the same as the process of obtaining information gain. The difference between them is the calculation method of entropy. In derivative of information gain, the entropy is calculated as

(28) e(D)=−pclog⁡pc−(1−pc)log⁡(1−pc)

where c is the class label of the time series which the candidate shapelet originated from, and pc is the probability of class c.

Coverage ratio

Based on the clustering technique, Yang & Jing (2024) proposed the coverage ratio to asses the shapelet quality. There are mainly four steps to obtain the coverage ratio of one candidate shapelet S: 1. Calculate the distances between each time series T and S.

2. Cluster the distance into C (which is the number of classes) groups, and numbered in ascending order of distance values. Thus, we obtained C groups ( G1, G2, ⋯, GC) with distances from small to large.

3. Count the number of time series with the same label as S in each group, and record them as N1, N2, ⋯, NC).

4. The final coverage ratio of S is calculated as (29) CR=N1|DL(S)|−λ∗∑i=2CNi|D|−|DL(S)|

where |D| is the number of all time series, |DL(S)| is the number of time series which has the same label as S, and λ∈(0,1] is the weight coefficient.

Combination of F1-score and AUC

Based on the one-vs-one encoding scheme, a multi-class classification problem can be converted into several binary-class problems (Bostrom & Bagnall, 2015). However, directly using binary-class measures, such as AUC, is not suitable for effectively determining the minority class. To address this issue, Yan & Cao (2020) combined the F1-score and AUC to evaluate the quality of candidate shapelets in the multi-class context, as shown in Eq. (30). In Eq. (30), F1 represents the F1-score, which can be calculated using Eq. (12);AUC can be obtained from Eq. (17); and IR refers to the imbalance ratio, computed as per Eq. (31). In Eq. (31), C denotes the number of classes, n is the total number of time series, and ni is the number of time series in class i.

(30) score=1IR⋅F1-(1−1IR)⋅AUC

(31) IR=1C∑i=1Cn−niC⋅ni.

Measures for ordinal classification

In time series classification, a natural order exists among the classes (Jing & Yang, 2024). In other words, misclassification costs are fundamentally asymmetric during ordinal classification, with differential penalties imposed based on class relationships. For instance, misclassifying a mild-case patient as severe should incur substantially higher penalties than misclassifying them as moderate (Guijo-Rubio et al., 2020). Unlike general measures, measures for ordinal classification must explicitly account for inter-class dependencies. This category of measures includes ordinal Fisher score, Spearman’s correlation coefficient, Pearson’s correlation coefficient, and CD-Cover.

Ordinal Fisher score

The Fisher score (Hart, Stork & Wiley, 2001) has been utilized for feature selection by several researchers (Gu, Li & Han, 2011). By reformulating it, Pérez-Ortiz et al. (2016) introduced the ordinal Fisher score to address the challenges of ordinal classification problems. Subsequently, Guijo-Rubio et al. (2020) applied the ordinal Fisher score to evaluate shapelet quality for ordinal time series classification problems. The process of obtaining the ordinal Fisher score involves four main steps: 1. Calculate the distances between each time series in the dataset and the candidate shapelet S, resulting in a list of distances L=⟨d1,d2,…,dn⟩.

2. Separate L into several lists L1,…,Li,…,LC based on class membership, such that the distances corresponding to class i are grouped into Li.

3. For each list Li, compute its mean Li¯ and its standard deviation Si.

4. Obtain the ordinal Fisher score as follows: (32) OFS=∑i=1C∑j=1C|i-j|(Li¯−Lj¯)2(C−1)∑i=1CSi2

where C is the number of classes. In Eq. (32), |i-j| serves to penalize the distances between farther classes.

Pearson’s correlation coefficient

Guijo-Rubio et al. (2020) adopted the Pearson’s correlation coefficient as the quality measure for ordinal classification. They calculated the Pearson’s correlation coefficient based on the distances obtained from the shapelet and the class labels. The Pearson’s correlation coefficient was primarily obtained through the following three steps: 1. Calculate the distances between each time series and the candidate shapelet S, recording them as L=⟨d1,d2,…,dn⟩.

2. Calculate the difference between the class of each time series and S, recording them as LC=⟨dc1,dc2,…,dcn⟩. For a given time series Ti and S, dci can be obtained as (33) dci=|PL(Ti)−PL(S)|

where L(Ti) is the class label of Ti, L(S) is the class label of S, and P∗ denotes the position of the class ∗ in the ordinal scale.

3. Finally, obtain the Pearson’s correlation coefficient as (34) PCC=cov(L,LC)SL⋅SLC

where cov(L,LC) is the covariance between L and LC, SL is the standard deviation of L, and SLC is the standard deviation of LC.

Spearman’s correlation coefficient

Spearman’s correlation coefficient was introduced by Guijo-Rubio et al. (2020) to assess the quality of the candidate shapelets. They mainly obtained Spearman’s correlation coefficient through the following five steps: 1. The distances between each time series and the candidate shapelet S were calculated and recorded as L=⟨d1,⋯,di,⋯,dn⟩.

2. Each distance di is assigned a rank value Ri from smallest to largest.

3. The difference between the class of each time series and S was calculated and recorded as LC=⟨dc1,⋯,dci,⋯,dcn⟩. Here, dci is calculated according to Eq. (33).

4. Each difference dci is assigned a rank value Rci from smallest to largest.

5. Finally, the Spearman’s correlation coefficient is calculated as (35) SCC=1−6∑j=1C(Ri−Rci)2n(n2−1)

where n is the number of time series.

Concentration and dominance of coverage

Jing & Yang (2024) proposed the CD-Cover as a quality measure for ordinal classification problems. Unlike most methods that utilize real subsequences, they ultimately selected some SAX (Lin et al., 2007) sequences as the final shapelets. The CD-Cover of candidate shapelet S can be obtained through the following six steps: 1. Represent all time series in the SAX format.

2. Count the coverage of S and denote it as ϕ=⟨λ1,⋯,λi,⋯,λC⟩, where C is the number of classes and λi is the number of time series containing S in class i.

3. Calculate the concentration of coverage of S as: (36) con=1−var(ϕ)(C−1)2/4

where var(ϕ) is the variance of the coverage ϕ.

4. Compute the coverage rate of S, denoting it as π=⟨κ1,⋯,κi,⋯,κC⟩, where κi is calculated as λi/ni ( ni is the number of time series in class i).

5. Calculate the dominance of coverage as: (37) dom=max1(π)−max2(π)

where max1(π) and max2(π) represent the largest and second-largest values in π, respectively.

6. Obtain the CD-Cover of S as: (38) σ=α⋅con+(1-α)⋅dom

where α∈(0,1) is a weight coefficient; in general, α is set to 0.5.

Discussion

The measures described in the preceding four sections are summarizes in Table 2. In addition to listing each measure and its corresponding category, Table 2 summarizes the shapelet measurement methods described in the preceding four chapters. In addition to listing each method and its corresponding category, the table also details each method’s applicable domains, indicates whether distance calculation between time series and subsequences is required during measurement, and specifies whether candidates are assessed individually or all at once.

Table 2 Shapelet quality measures summarized.

Category	Measure	The proposed paper	Usage domain	Distance calculation?	Assess methods	
General measures	Information gain	Ye & Keogh (2009)	All applicable	Yes	Individually	
	F-statistic	Lines et al. (2012)		Yes	Individually	
	Kruskal-Wallis	Lines & Bagnall (2012)		Yes	Individually	
	Mood’s median	Lines & Bagnall (2012)		Yes	Individually	
	F-measure	Xing et al. (2011)		Yes	Individually	
	Silhouette score			Yes	Individually	
	Cumulative distance	Liu et al. (2016)		Yes	Individually	
	Importance measure	Ji et al. (2022)		Yes	All at once	
Measures for binary-class	AUC	Yan & Cao (2020)	Binary-class task	Yes	Individually	
	Distance measure	Chen & Wan (2023)		Yes	Individually	
	Location measure	Chen & Wan (2023)		Yes	Individually	
	Chi-Square statistic	Bock et al. (2018)		Yes	Individually	
	Gap	Ulanova, Begum & Keogh (2015)		Yes	Individually	
Measure for multi-class	Derivative of information gain	Bostrom & Bagnall (2015)	Multi-class task	Yes	Individually	
	Coverage ratio	Yang & Jing (2024)		Yes	Individually	
	Combination of F1-score and AUC	Yan & Cao (2020)		Yes	Individually	
Measures for ordinal classification	Ordinal Fisher score	Guijo-Rubio et al. (2020)	Ordinal classification	Yes	Individually	
	Pearson’s correlation coefficient	Guijo-Rubio et al. (2020)		Yes	Individually	
	Spearman’s correlation coefficient	Guijo-Rubio et al. (2020)		Yes	Individually	
	CD-Cover	Jing & Yang (2024)		No	Individually	

The primary advantage of general measures is their broad applicability across all fields. Information gain, the earliest method in this category applied to evaluate shapelet candidates, was central to the initial proposal of shapelet-based time series classification (Ye & Keogh, 2009) and remains the most widely used measure today. However, Information gain suffers from high computational complexity (Bostrom & Bagnall, 2017). A key reason to this complexity is its requirement to find the optimal distance threshold that splits the training data into two distinct subsets. This process involves sorting the distances from all time series to a candidate shapelet and then evaluating potential thresholds, typically at least the midpoints between adjacent sorted distances (Charane, Ceccarello & Gamper, 2024). Measures such as F-statistic, Kruskal-Wallis, Mood’s median, Silhouette score, F-measure, and cumulative distance each circumvent the threshold search step in different ways. While these alternatives all achieve some speedup compared to IG, the improvement is not orders of magnitude (Hills et al., 2014). One significant reason is that they still evaluate candidates individually. To address this limitation, importance measure leveraging random forests were introduced, enabling the simultaneous evaluation of all candidates (Ji et al., 2022). Despite their general applicability to time series classification problems, these ggeneral measures may fail to deliver optimal performance for certain specialized problems due to their specific characteristics.

In binary-class problems, the outcomes are exclusively positive or negative, and class imbalance is common. The AUC effectively addresses this imbalance, offering the advantage of being insensitive to data distribution (Yan & Cao, 2020). Distance measures and location measures assess shapelets based on inter-class and intra-class differences. While distance measures focus primarily on dissimilarity magnitudes, location measures emphasize the positions where similar subsequences occur (Chen & Wan, 2023). Combining these measures can yield superior results. To better distinguish inter-class and intra-class differences, the Chi-Square statistic employs the χ2 test to evaluate statistical significance (Bock et al., 2018). Gap, initially applied to clustering problems (Ulanova, Begum & Keogh, 2015), was later adapted for binary-class problems. Compared to the previously mentioned measures, Gap’s most significant feature is its incorporation of both the mean distance and the standard deviation of distances within each class. Measures designed for binary-class problems require the one-vs.-all strategy to be applied when dealing with multi-class problems. The one-vs.-all approach decomposes the multi-class task into multiple binary classification tasks.

Unlike general measures which assess how well a single shapelet candidate separates all classes, measures for multi-class focus on a candidate’s discriminative power for a specific class (Bostrom & Bagnall, 2015). Consequently, the derivative of information gain modifies the entropy calculation formula. An advantage of the derivative of information gain is that, apart from its entropy computation, its other steps align with general measures, enhancing its interpretability (Bostrom & Bagnall, 2015). The combination of F1-score and AUC extends the AUC concept from binary classification to multi-class problems. This measure retains AUC’s key advantage of insensitivity to class distribution (Yan & Cao, 2020). The coverage ratio assesses the discriminative power of shapelet candidates by revealing the cover relationship between a candidate and time series through clustering their distances (Yang & Jing, 2024). Measures in this category translate global data partitioning into a candidate’s discriminative ability for a specific class. Compared to general measures, their primary drawback is the potential risk of compromising overall classification accuracy.

The most significant feature of measures for ordinal classification is that these measure explicitly account for inter-class dependencies. In order to obtain shapelets minimising the most severe errors in the ordinal scale, Ordinal Fisher score, Spearman’s correlation coefficient, and Pearson’s correlation coefficient adopted the difference between class labelsto penalize the distances between farther classes (Guijo-Rubio et al., 2020). By doing so, the farther the relationship between classes, the greater the punishment. Among them, the Ordinal Fisher score only evaluating the separability obtained in accordance to the ordinal scale. Whist, Spearman’s correlation coefficient and Pearson’s correlation coefficient take into account the category of the shapelet being evaluated (Guijo-Rubio et al., 2020). The most significant feature of CD-Cover is that it does not require calculating the distance between the shapelet candidate and the time series (Jing & Yang, 2024). This greatly improves the efficiency of shapelet measurement. This type of measures needs to consider the relationships among classes. When there is no relationship among classes, weight design will be meaningless. This greatly limits the applicability of these measures.

The defining feature of measures for ordinal classification is their explicit incorporation of inter-class dependencies. To identify shapelets minimizing severe errors across the ordinal scale, methods like Ordinal Fisher score, Spearman’s correlation coefficient, and Pearson’s correlation coefficient leverage class label differences to penalize distances between distantly ranked classes (Guijo-Rubio et al., 2020). This penalty scales with class separation: greater ordinal distance results in stronger penalization. Among these, Ordinal Fisher score focuses exclusively on ordinal separability, while Spearman’s and Pearson’s coefficients additionally account for the evaluated shapelet’s category (Guijo-Rubio et al., 2020). CD-Cover’s core innovation lies in eliminating distance calculations between shapelet candidates and time series (Jing & Yang, 2024), significantly enhancing measurement efficiency. A critical limitation of these ordinal measures is their dependence on meaningful class relationships. When classes lack inherent ordering, the weight-design mechanisms become invalid, restricting their applicability to inherently ordinal problems.

From the above description, we can see that the main difference between the four types of methods lies in their applicable fields. In addition, the final classification accuracy performance based on measures in the same category is basically on the same level. Although there are slight differences in the efficiency of measures within the same category, except for importance measure and CD-Cover, other methods are basically of the same order of magnitude.

Challenges

After a thorough review of shapelet quality measures, we identify the following points: shapelets selected through these measures can accurately classify time series data and provide a degree of interpretability. Despite the promising results, several challenges remain: Excessive dependence on Euclidean distance. With the exception of CD-Cover, all measures necessitate the computation of distances between candidate shapelets and time series data. The predominant reliance on Euclidean distance in the existing literature (Costa et al., 2020; Amouri et al., 2022) raises concerns about its limitations in capturing complex relationships within diverse datasets. This dependence may compromise performance when the underlying data distributions deviate from the assumptions inherent in Euclidean metrics.

High time complexity. Many shapelet quality measures exhibit high computational intensity (Wan et al., 2023). This complexity arises not only from the extensive number of candidate shapelets (Ji et al., 2019) but also from the inherently time-consuming nature of the measures themselves (Ye & Keogh, 2011; Mbouopda & Nguifo, 2024). The trade-off between rigorous evaluation and practical applicability becomes evident; while thorough assessments are essential for robust model performance, they can significantly hinder scalability and real-time processing capabilities.

Lack of consideration for interpretability. One notable advantage of shapelets is their intuitive interpretability (Chen & Wan, 2023). However, current measures tend to emphasize categorization ability at the expense of comprehensibility for end-users. This oversight presents a critical trade-off: while performance metrics may indicate efficacy in classification tasks, they often neglect the importance of presenting understandable results that can be effectively communicated to users. Balancing interpretability with performance remains a vital consideration for advancing shapelet-based methods.

Future research directions

In response to the above challenges, combined with current research hotspots, several promising areas for further research have been identified, as discussed below: Revolutionizing measure efficiency for next-generation challenges. As datasets scale toward exabyte volumes and real-time analytics demands intensify, future research must transcend hardware-specific optimizations. We envision algorithmic-level breakthroughs: measures leveraging quantum-inspired annealing for hyper-fast candidate screening, or federated evaluation schemes preserving privacy in distributed environments. Simultaneously, rethinking distance-free quality assessment—potentially through topological invariants or symbolic representations—could bypass computational bottlenecks entirely.

Embedding human-centric interpretability into quality assessment. The quest for intuitive shapelets demands deeper collaboration between ML and human-computer interaction research. Future measures could incorporate neurosymbolic frameworks, where quality scores reflect both statistical robustness and alignment with domain experts’ mental models. Integrating knowledge graphs for semantic validation, or designing generative adversarial networksto synthesize “interpretability audits”, might bridge the gap between technical efficacy and actionable insights.

Convergence of shapelets and deep learning architectures. With shapelets evolving into trainable convolutional filters, quality measures must advance beyond traditional criteria. Future work should develop latent-space quality metrics that evaluate shapelets’ contributions to model disentanglement, adversarial robustness, or multi-scale feature coherence. Exploring connections to information bottleneck theory could provide theoretical foundations for “learnable shapelet” assessment, positioning shapelets as interpretable building blocks within foundation models.

Shapelets in the era of large language and multimodal models. The nascent integration of shapelets with large language models opens transformative pathways. Research should explore cross-modal quality measures evaluating how shapelets ground temporal reasoning in language models (e.g., via attention-map analysis), or facilitate multimodal alignment in video-text systems. As large language models absorb time-series data, developing “prompt-aware” shapelet metrics that optimize for human-artificial intelligence (AI) collaboration will be critical for scientific discovery and industrial.

Conclusion

In this paper, we provide a comprehensive review of shapelet quality measures. The key contributions of this work are summarized as follows: A taxonomy of shapelet quality measures, specifically from the perspective of classification tasks, is proposed. The measures are categorized into four main groups: general measures, measures for binary-class, measures for multi-class, and measures for ordinal classification. We present a detailed description of the current shapelet quality measures, allowing users to gain a fundamental understanding of the various methods. Additionally, we offer a summary of the four categories of measures and discuss the advantages and disadvantages of each. Based on these discussions, we outline the challenges faced by current research and propose directions for future investigations. We hope this survey will serve as a valuable resource for researchers in the field.

Based on our research, we anticipate that future hotspots in shapelet measurement will focus on several key areas, including improving efficiency, enhancing interpretability, leveraging artificial intelligence technologies, and integrating large-scale models. These areas represent an expansion of the current research focus, offering broader opportunities for innovation and application in the field.

The author extends thanks to DeepSeek-R1 (DeepSeek) for its professional language polishing services. Its AI-powered proofreading contributed substantially to enhancing the fluency and grammatical accuracy of this article.

Additional Information and Declarations

Competing Interests

The authors declare that they have no competing interests.

Author Contributions

Teng Li conceived and designed the experiments, performed the experiments, analyzed the data, prepared figures and/or tables, authored or reviewed drafts of the article, and approved the final draft.

Xiaodong Guo conceived and designed the experiments, analyzed the data, authored or reviewed drafts of the article, and approved the final draft.

Cun Ji conceived and designed the experiments, performed the experiments, analyzed the data, prepared figures and/or tables, authored or reviewed drafts of the article, and approved the final draft.

Data Availability

The following information was supplied regarding data availability:

This article is a literature review and did not generate raw data.

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
