# Peer review of "A literature survey of shapelet quality measures for time series classification"

_PeerJ Computer Science, doi:10.7717/peerj-cs.3115_

## Round 0.1 · original submission · Major Revisions

**Language Note:** The review process has identified that the English language must be improved. PeerJ can provide language editing services - please contact us at [email protected] for pricing (be sure to provide your manuscript number and title). Alternatively, you should make your own arrangements to improve the language quality and provide details in your response letter. – PeerJ Staff

Reviewer 1 ·

Basic reporting

This is a very complete survey of shapelet quality measures.

It is a little uncritical, but it could be a useful resource for someone working in this area.

One area for “Future Research Directions” listed is “Combination measures with non-Euclidean distances,” and DTW is suggested. However, as the length of the shapelets is generally much less than the length of the full time series to be classified, the difference between DTW and ED tends to zero. In other words, using DTW does not seem to help for shapelets.

Experimental design

The literature survey is boilerplate (but perhaps that is a good thing)

Validity of the findings

No issues here

Additional comments

Figure 1 is a bitmap, Please use vector graphics.
If you look at the bottom blue arrow, it is not straight (use “snap to grid”)

Time series are among the most visually intuitive types of data.
And shapelets are proposed as being visually intuitive.
As such, it is disappointing that we don’t see any plots of time series/shapelets.

Cite this review as

Reviewer 2 ·

Basic reporting

The review is generally well-written; however, the different sections are not well connected
and read more like a list of information than a coherent, unified discourse.
The topic of the review is well introduced, however the description of the importance of the
field is too broad and requires more specific details about the importance of Shapelets based
methods in Time series analysis. Additionally, the explanation of basic elements (such as the
definition of a Shapelets), fundamental to fully understand the rest of the paper, is not clear
in the introduction (lines 35-44). The introduction would also require more a detailed
contextualization of this work in the current literature.
It is unclear whether similar works exist. Although the authors state that no prior work has
pursued the same objective, the paper would benefit from a discussion of related studies,
whether similar or different.

Experimental design

The investigation is well executed, and the methodology is described with sufficient detail.
The topic is covered thoroughly, with various categories of approaches explored.
The sources are adequately cited.
The organization follows a logical and coherent structure. However, at times, the sections
feel disconnected from one another, which makes the reading somewhat difficult

Validity of the findings

The impact and novelty of the research are not sufficiently discussed. The review would
benefit from better contextualization within the current state of the art.
The rationale behind the work is unclear. In particular, the introduction and conclusions are
not well connected and fail to emphasize the importance of the study.
The paper would benefit from a clearer separation between the discussion and conclusion
sections. In the discussion, the authors should analyze the advantages and disadvantages
of current methods and consider outlining future research directions from a broader
perspective than the one currently adopted.
The conclusion should summarize the work and, if appropriate, highlight some potential
areas for future research.
Although the conclusion identifies gaps and future directions, the discussion is not
sufficiently developed.

Additional comments

The problem investigated and the research methodology are both valid.
However, the paper has some drawbacks: it lacks adequate discussion and contextualization
of the problem within the current state of the art. Additionally, it does not sufficiently explain
the foundational knowledge required to fully understand the content.
The paper is well structured and the English is generally good. However, the sections are not
well connected to each other.
It is important to separate the discussion from the conclusions and to expand the discussion
section—for example, by addressing the pros and cons of different metrics

Cite this review as

Reviewer 3 ·

Basic reporting

1. There are several grammatical mistakes and awkward phrasing that make sentences less clear (e.g., “papers were filtered out,” “an officially published paper,” and “each shapelet quality measure will be described in detail”).
2. Some sentences, such as “Only 93 papers that formally describe the method of measuring shapelet quality were ultimately selected,” need to be refined, such as (e.g., “Only 93 papers that formally described methods for evaluating shapelet quality were ultimately selected”).
3. There is a need for clarity and consistency in the use of visual aids, such as Figures 1 and 2, and for improved readability in Table 1.
4. Consider consolidating if possible, as some inline citations, such as "Ji et al., 2022, 2023," are repetitive.

Experimental design

5. Clarify the recentness of the review—how does it compare with other surveys or reviews post-2020?
6. The authors must provide a justification for limiting their search to Google Scholar. The need to make it more comprehensive by considering adding IEEE, ACM Digital Library, Scopus, or Web of Science.
7. Authors need to clarify how potential biases were mitigated in selecting and interpreting the 93 final papers.

Validity of the findings

8. The “Challenges” and “Future Directions” sections need to be improved in terms of critical discussion (e.g., trade-offs between interpretability and performance, dataset diversity, etc.).
9. Authors need to consider investigating real-world domains where specific shapelet measures have superior performance (e.g., healthcare, finance).

Cite this review as

---

## Round 0.2 · accepted · Accept

Dear authors, we are pleased to verify that you meet the reviewer's valuable feedback to improve your research.

Thank you for considering PeerJ Computer Science and submitting your work.

Kind regards
PCoelho

Reviewer 1 ·

Basic reporting

I am happy with the revised work (except figures)

Experimental design

-

Validity of the findings

-

Cite this review as

Reviewer 2 ·

Basic reporting

The current version of the paper has been improved, and the authors have adequately and satisfactorily addressed all the issues raised in the previous review.

Experimental design

-

Validity of the findings

-

Cite this review as